# Optimal Algorithms for Continuous Non-monotone Submodular and DR-Submodular Maximization

**Rad Niazadeh**[*]
Department of Computer Science
Stanford University, Stanford, CA 95130
rad@cs.stanford.edu

Tim Roughgarden[†]
Department of Computer Science
Stanford University, Stanford, CA 95130
tim@cs.stanford.edu

Joshua R. Wang
Google, Mountain View, CA 94043
joshuawang@google.com

## Abstract

In this paper we study the fundamental problems of maximizing abcontinuous non-monotone submodular function over a hypercube, with and without coordinate-wise concavity. This family of optimization problems has several applications in machine learning, economics, and communication systems. Our main result is the first $\frac{1}{2}$-approximation algorithm for continuous submodular function maximization; the approximation factor of $\frac{1}{2}$ is the best possible for algorithms that use only polynomially many queries. For the special case of DR-submodular maximization, i.e., when the submodular functions is also coordinate-wise concave along all coordinates, we provide a faster $\frac{1}{2}$-approximation algorithm that runs in almost linear time. Both of these results improve upon prior work [Bian et al., 2017a,b, Soma and Yoshida, 2017, Buchbinder et al., 2012, 2015].

Our first algorithm is a single-pass algorithm that uses novel ideas such as reducing the guaranteed approximation problem to analyzing a zero-sum game for each coordinate, and incorporates the geometry of this zero-sum game to fix the value at this coordinate. Our second algorithm is a faster single-pass algorithm that exploits coordinate-wise concavity to identify a monotone equilibrium condition sufficient for getting the required approximation guarantee, and hunts for the equilibrium point using binary search. We further run experiments to verify the performance of our proposed algorithms in related machine learning applications.

## 1 Introduction

*Submodular optimization* is a sweet spot between tractability and expressiveness, with numerous applications in machine learning (e.g., Krause and Golovin [2014], and see below) and with many algorithms that are both practical and enjoy good rigorous guarantees (e.g., Buchbinder et al. [2012, 2015]). In general, a real-valued function $\mathcal{F}$ defined on a lattice $\mathcal{L}$ is *submodular* if and only if

$$\mathcal{F}(x \vee y) + \mathcal{F}(x \wedge y) \leq \mathcal{F}(x) + \mathcal{F}(y)$$

for all $x, y \in \mathcal{L}$, where $x \vee y$ and $x \wedge y$ denote the join and meet, respectively, of $x$ and $y$ in the lattice $\mathcal{L}$. Such functions are generally neither convex nor concave. In one of the most commonly studied examples, $\mathcal{L}$ is the lattice of subsets of a fixed ground set (or a sublattice thereof), with union and intersection playing the roles of join and meet, respectively.

---

[*]Rad Niazadeh was supported by Stanford Computer Science Motwani Fellowship.

[†]Tim Roughgarden was supported in part by Google Faculty Grant, Guggenheim Fellowship, and NSF Grants CCF-1524062 and CCF-1813188.

This paper concerns a different well-studied setting, where $\mathcal{L}$ is a hypercube (i.e., $[0,1]^n$), with componentwise maximum and minimum serving as the join and meet, respectively.[3] We consider the fundamental problem of (approximately) maximizing a continuous and nonnegative submodular function over the hypercube.[4] The function $\mathcal{F}$ is given as a "black box" and can only be accessed by querying its value at a point. We are interested in algorithms that use at most a polynomial (in $n$) number of queries. We do not assume that $\mathcal{F}$ is monotone (otherwise the problem is trivial).

We next briefly mention four applications of maximizing a non-monotone submodular function over a hypercube that are germane to machine learning and other related application domains.[5]

*Non-concave quadratic programming.* In this problem, the goal is to maximize $\mathcal{F}(\mathbf{x}) = \frac{1}{2}\mathbf{x}^T\mathbf{H}\mathbf{x} + \mathbf{h}^T\mathbf{x} + c$, where the off-diagonal entries of $\mathbf{H}$ are non-positive. One application of this problem is to large-scale price optimization on the basis of demand forecasting models [Ito and Fujimaki, 2016].

*Map inference for Determinantal Point Processes (DPP).* DPPs are elegant probabilistic models that arise in statistical physics and random matrix theory. DPPs can be used as generative models in applications such as text summarization, human pose estimation, and news threading tasks [Kulesza et al., 2012]. The approach in Gillenwater et al. [2012] to the problem boils down to maximize a suitable submodular function over the hypercube, accompanied with an appropriate rounding (see also [Bian et al., 2017a]). One can also think of regularizing this objective function with $\ell_2$-norm regularizer, to avoid overfitting, and the function will still remain submodular.

*Log-submodularity and mean-field inference.* Another probabilistic model that generalizes DPPs and all other strong Rayleigh measures [Li et al., 2016, Zhang et al., 2015] is the class of *log-submodular* distributions over sets, i.e., $p(S) \sim \exp(\mathcal{F}(S))$ where $\mathcal{F}(\cdot)$ is a set submodular function. MAP inference over this distribution has applications in machine learning [Djolonga and Krause, 2014]. One variational approach towards this MAP inference task is to use *mean-field inference* to approximate the distribution $p$ with a product distribution $\mathbf{x} \in [0,1]^n$, which again boils down to submodular function maximization over the hypercube (see [Bian et al., 2017a]).

*Revenue maximization over social networks.* Here, there is a seller who wants to sell a product over a social network of buyers. To do so, it freely assigns trial products and fractions thereof to the buyers in the network [Bian et al., 2017b, Hartline et al., 2008]. For this problem, one can reduce it to maximizing an objective function that takes into account two parts: the revenue gain from those who did not get a free product, where the revenue function for any such buyer is a non-negative non-decreasing and submodular function $R_i(\mathbf{x})$; and the revenue loss from those who received the free product, where the revenue function for any such buyer is a non-positive non-increasing and submodular function $\bar{R}_i(\mathbf{x})$. The combination for all buyers is a non-monotone submodular function. It also is non-negative at $\vec{0}$ and $\vec{1}$, by extending the model and accounting for extra revenue gains from buyers with free trials.

**Our Results.**  Maximizing a submodular function over the hypercube is at least as difficult as over the subsets of a ground set.[6] For the latter problem, the best approximation ratio achievable by an algorithm making a polynomial number of queries is $\frac{1}{2}$; the (information-theoretic) lower bound is due to [Feige et al., 2007, 2011], the optimal algorithm to [Buchbinder et al., 2012, 2015]. Thus, the best-case scenario for maximizing a submodular function over the hypercube (using polynomially many queries) is a $\frac{1}{2}$-approximation. The main result of this paper achieves this best-case scenario:

> *There is an algorithm for maximizing a continuous submodular function over the hypercube that guarantees a $\frac{1}{2}$-approximation while using only a polynomial number of queries to the function under mild continuity assumptions.*

Our algorithm is inspired by the *bi-greedy* algorithm of Buchbinder et al. [2015], which maximizes a submodular set function; it maintains two solutions initialized at $\vec{0}$ and $\vec{1}$, goes over coordinates

sequentially, and makes the two solutions agree on each coordinate. The algorithmic question here is how to choose the new coordinate value for the two solutions, so that the algorithm gains enough value relative to the optimum in each iteration. Prior to our work, the best-known result was a $\frac{1}{3}$-approximation [Bian et al., 2017b], which generalized the simple non-optimal $\frac{1}{3}$-approximation deterministic bi-greedy algorithm of [Buchbinder et al., 2012, 2015] for set functions to continuous domains. However, to get the optimal approximation factor and systematically passing the barrier of pure continuous submodularity, our algorithm requires a number of new ideas, including a reduction to the analysis of a zero-sum game for each coordinate, and the use of the special geometry of this game to bound the value of the game at its equilibrium. See Section 2 for more details.

The second and third applications above induce objective functions that, in addition to being submodular, are concave in each coordinate. [7] This class of functions is called *DR-submodular* in the literature (e.g., in [Soma and Yoshida, 2015] and based on diminishing returns defined in [Kapralov et al., 2013]). Here, an optimal $\frac{1}{2}$-approximation algorithm was recently already known on integer lattices [Soma and Yoshida, 2017], that can easily be generalized to our continuous setting as well; our contribution is a significantly faster such bi-greedy algorithm. The main idea here is to identify a monotone equilibrium condition sufficient for getting the required approximation guarantee, which enables a binary search-type solution. See Section 3 for more details.

We also run experiments to verify the performance of our proposed algorithms in practical machine learning applications. We observe that our algorithms match the performance of the prior work, while providing either a better guaranteed approximation or a better running time.

**Further Related Work.** Buchbinder and Feldman [2016] derandomize the bi-greedy algorithm. Staib and Jegelka [2017] apply continuous submodular optimization to budget allocation, and develop a new submodular optimization algorithm to this end. Hassani et al. [2017] give a $\frac{1}{2}$-approximation for *monotone* continuous DR-submodular functions under convex constraints, which is later improved to $(1 - \frac{1}{e})$-approximation in Mokhtari et al. [2018] (even for stochastic functions). Gotovos et al. [2015] consider (adaptive) submodular maximization when feedback is given after an element is chosen. Chen et al. [2018], Roughgarden and Wang [2018] consider submodular maximization in the context of online no-regret learning. Mirzasoleiman et al. [2013] show how to perform submodular maximization with distributed computation. Submodular minimization is studied in Schrijver [2000], Iwata et al. [2001]. See Bach et al. [2013] for a survey on more applications in machine learning.

**Variations of Continuous Submodularity.** We consider non-monotone non-negative *continuous submodular functions*, i.e., $\mathcal{F} : [0,1] \to [0,1]^n$ s.t. $\forall \mathbf{x}, \mathbf{y} \in [0,1]^n$, $\mathcal{F}(\mathbf{x}) + \mathcal{F}(\mathbf{y}) \geq \mathcal{F}(\mathbf{x} \vee \mathbf{y}) + \mathcal{F}(\mathbf{x} \wedge \mathbf{y})$, where $\vee$ and $\wedge$ are coordinate-wise max and min operations. Two related properties are *weak Diminishing Returns Submodularity* (weak DR-SM) and *strong Diminishing Returns Submodularity* (strong DR-SM) [Bian et al., 2017b], formally defined below. Indeed, weak DR-SM is equivalent to submodularity (see Proposition 3 in the supplement), and hence we use these terms interchangeably.

**Definition 1** (Weak/Strong DR-SM). Consider a continuous function $\mathcal{F} : [0,1]^n \to [0,1]$:

- Weak DR-SM (continuous submodular): $\forall i \in [n]$, $\forall \mathbf{x}_{-i} \leq \mathbf{y}_{-i} \in [0,1]^n$, and $\forall \delta \geq 0, \forall z$

$$\mathcal{F}(z + \delta, \mathbf{x}_{-i}) - \mathcal{F}(z, \mathbf{x}_{-i}) \geq \mathcal{F}(z + \delta, \mathbf{y}_{-i}) - \mathcal{F}(z, \mathbf{y}_{-i})$$

- Strong DR-SM (*DR-submodular* ): $\forall i \in [n]$, $\forall \mathbf{x} \leq \mathbf{y} \in [0,1]^n$, and $\forall \delta \geq 0$:

$$\mathcal{F}(x_i + \delta, \mathbf{x}_{-i}) - \mathcal{F}(\mathbf{x}) \geq \mathcal{F}(y_i + \delta, \mathbf{y}_{-i}) - \mathcal{F}(\mathbf{y})$$

As simple corollaries, a twice-differentiable $\mathcal{F}$ is strong DR-SM if and only if all the entries of its Hessian are non-positive, and weak DR-SM if and only if all of the *off-diagonal* entries of its Hessian are non-positive. Also, weak DR-SM together with concavity along each coordinate is equivalent to strong DR-SM (see Proposition 3 in the supplementary materials for more details).

**Coordinate-wise Lipschitz Continuity.** Consider univariate functions generated by fixing all but one of the coordinates of the original function $\mathcal{F}(\cdot)$. In future sections, we sometimes require mild technical assumptions on the Lipschitz continuity of these single dimensional functions.

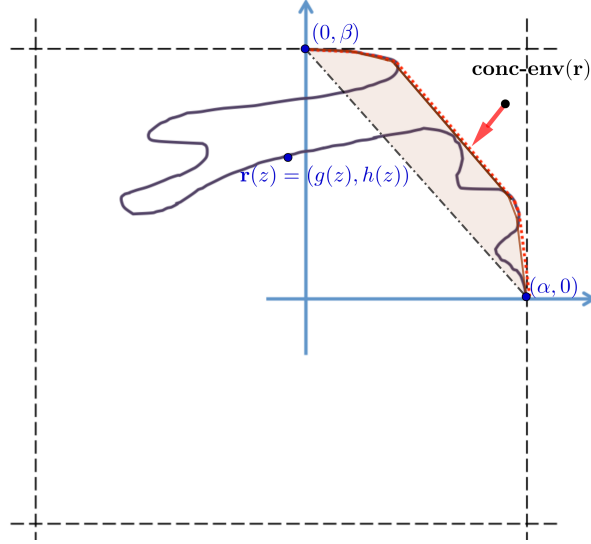

Figure 1: Continuous curve $\mathbf{r}(z)$ in $\mathbb{R}^2$ (dark blue), positive-orthant concave envelope (red).

**Definition 2** (Coordinate-wise Lipschitz). *A function $\mathcal{F} : [0,1]^n \to [0,1]$ is coordinate-wise Lipschitz continuous if there exists a constant $C > 0$ such that $\forall i \in [n], \forall \mathbf{x}_{-i} \in [0,1]^n$, the single variate function $\mathcal{F}(\cdot, \mathbf{x}_{-i})$ is $C$-Lipschitz continuous, i.e.,*

$$\forall z_1, z_2 \in [0,1] : \quad |\mathcal{F}(z_1, \mathbf{x}_{-i}) - \mathcal{F}(z_2, \mathbf{x}_{-i})| \leq C|z_1 - z_2|$$

## 2 Weak DR-SM Maximization: Continuous Randomized Bi-Greedy

Our first main result is a $\frac{1}{2}$-approximation algorithm (up to additive error $\delta$) for maximizing a continuous submodular function $\mathcal{F}$, a.k.a. weak DR-SM, which is information-theoretically optimal [Feige et al., 2007, 2011]. This result assumes that $\mathcal{F}$ is coordinate-wise Lipschitz continuous.[8] Before describing our algorithm, we introduce the notion of the *positive-orthant concave envelope* of a two-dimensional curve, which is useful for understanding our algorithm.

**Definition 3.** Consider a curve $\mathbf{r}(z) = (g(z), h(z)) \in \mathbb{R}^2$ over the interval $z \in [Z_l, Z_u]$ such that:

1. $g : [Z_l, Z_u] \to [-1, \alpha]$ and $h : [Z_l, Z_u] \to [-1, \beta]$ are both continuous,

2. $g(Z_l) = h(Z_u) = 0$, and $h(Z_l) = \beta \in [0,1]$, $g(Z_u) = \alpha \in [0,1]$.

Then the *positive-orthant concave envelope* of $\mathbf{r}(\cdot)$, denoted by $\mathsf{conc\text{-}env}(\mathbf{r})$, is the smallest concave curve in the positive-orthant upper-bounding all the points $\{\mathbf{r}(z) : z \in [Z_l, Z_u]\}$ (see Figure 1), i.e.,

$$\mathsf{conc\text{-}env}(\mathbf{r}) \triangleq \text{upper-face}\left(\text{conv}\left(\{\mathbf{r}(z) : z \in [Z_l, Z_u]\}\right) \cap \left\{(g', h') \in [0,1]^2 : \frac{h'}{\beta} + \frac{g'}{\alpha} \geq 1\right\}\right)$$

We start by describing a vanilla version of our algorithm for maximizing $\mathcal{F}$ over the unit hypercube, termed as *continuous randomized bi-greedy* (Algorithm 1). This version assumes blackbox oracle access to algorithms for a few computations involving univariate functions of the form $\mathcal{F}(., \mathbf{x}_{-i})$ (e.g., maximization over $[0,1]$, computing $\mathsf{conc\text{-}env}(.)$, etc.). We first prove that the vanilla algorithm finds a solution with an objective value of at least $\frac{1}{2}$ of the optimum. In Section 2.2, we show how to approximately implement these oracles in polynomial time when $\mathcal{F}$ is coordinate-wise Lipschitz.

**Theorem 1.** *If $\mathcal{F}(\cdot)$ is non-negative and continuous submodular (or equivalently is weak DR-SM), then Algorithm 1 is a randomized $\frac{1}{2}$-approximation algorithm, i.e., returns $\hat{\mathbf{z}} \in [0,1]^n$ s.t.*

$$2\mathbf{E}\left[\mathcal{F}(\hat{\mathbf{z}})\right] \geq \mathcal{F}(\mathbf{x}^*), \qquad \text{where } \mathbf{x}^* \in \underset{\mathbf{x} \in [0,1]^n}{\text{argmax}} \, \mathcal{F}(\mathbf{x}) \text{ is the optimal solution.}$$

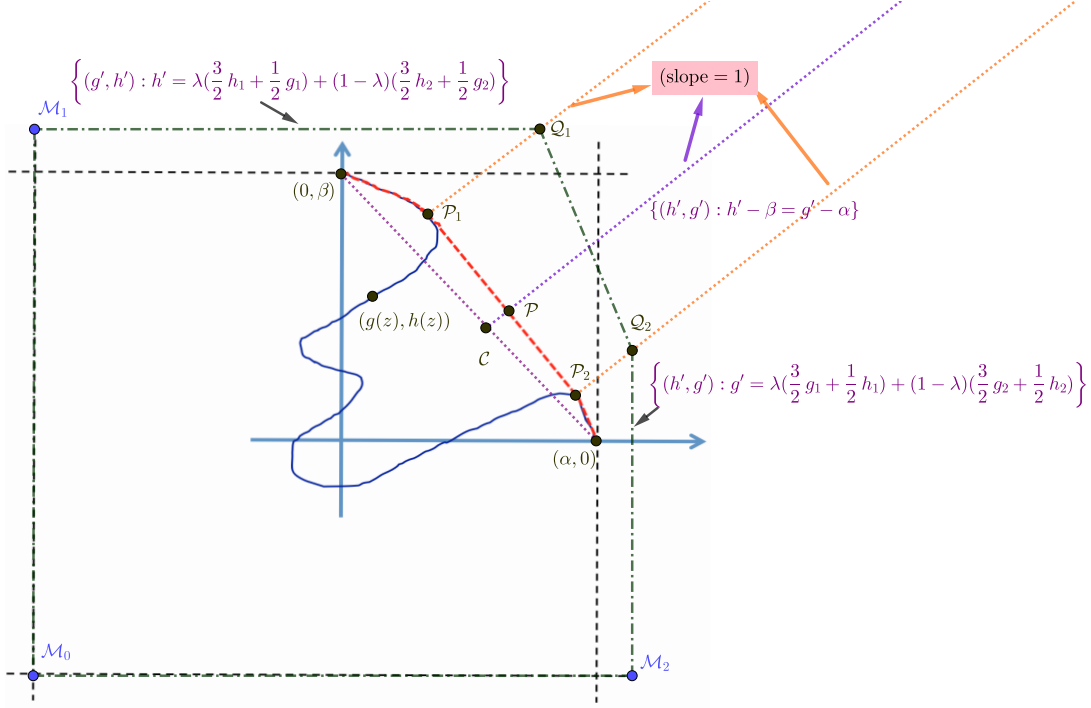

Figure 2: Pentagon $(\mathcal{M}_0, \mathcal{M}_1, \mathcal{Q}_1, \mathcal{Q}_2, \mathcal{M}_2)$= ADV player's positive region against a mixed strategy over two points $\mathcal{P}_1$ and $\mathcal{P}_2$.

## 2.1 Analysis of the Continuous Randomized Bi-Greedy (Proof of Theorem 1)

We start by defining these vectors, used in our analysis in the same spirit as Buchbinder et al. [2015]:

$$
\begin{aligned}
i \in [n]: \ \mathbf{X}^{(i)} &\triangleq (\hat{z}_1, \ldots, \hat{z}_i, 0, 0, \ldots, 0), & \mathbf{X}^{(0)} &\triangleq (0, \ldots, 0) \\
i \in [n]: \ \mathbf{Y}^{(i)} &\triangleq (\hat{z}_1, \ldots, \hat{z}_i, 1, 1, \ldots, 1), & \mathbf{Y}^{(0)} &\triangleq (1, \ldots, 1) \\
i \in [n]: \ \mathbf{O}^{(i)} &\triangleq (\hat{z}_1, \ldots, \hat{z}_i, x^*_{i+1}, \ldots, x^*_n), & \mathbf{O}^{(0)} &\triangleq (x^*_1, \ldots, x^*_n)
\end{aligned}
$$

Note that $\mathbf{X}^{(i)}$ and $\mathbf{Y}^{(i)}$ (or $\mathbf{X}^{(i-1)}$ and $\mathbf{Y}^{(i-1)}$) are the values of $\mathbf{X}$ and $\mathbf{Y}$ at the end of (or at the beginning of) the $i^{\text{th}}$ iteration of Algorithm 1. In the remainder of this section, we give the high-level proof ideas and present some proof sketches. See the supplementary materials for the formal proofs.

***Reduction to Coordinate-wise Zero-sum Games.*** For each coordinate $i \in [n]$, we consider a sub-problem. In particular, define a two-player *zero-sum game* played between the *algorithm player* (denoted by ALG) and the *adversary player* (denoted by ADV). ALG selects a (randomized) strategy $\hat{z}_i \in [0,1]$, and ADV selects a (randomized) strategy $x^*_i \in [0,1]$. Recall the descriptions of $g(z)$ and $h(z)$ at iteration $i$ of Algorithm 1,:

$$
g(z) = \mathcal{F}(z, \mathbf{X}^{(i-1)}_{-i}) - \mathcal{F}(Z_l, \mathbf{X}^{(i-1)}_{-i}) \ , \ \ h(z) = \mathcal{F}(z, \mathbf{Y}^{(i-1)}_{-i}) - \mathcal{F}(Z_u, \mathbf{Y}^{(i-1)}_{-i}).
$$

We now define the utility of ALG (negative of the utility of ADV) in our zero-sum game as follows:

$$
\mathcal{V}^{(i)}(\hat{z}_i, x^*_i) \triangleq \frac{1}{2} g(\hat{z}_i) + \frac{1}{2} h(\hat{z}_i) - \max\left(g(x^*_i) - g(\hat{z}_i), h(x^*_i) - h(\hat{z}_i)\right). \tag{1}
$$

Suppose the expected utility of ALG is non-negative at the equilibrium of this game. In particular, suppose ALG's randomized strategy $\hat{z}_i$ (in Algorithm 1) guarantees that for every strategy $x^*_i$ of ADV the expected utility of ALG is non-negative. If this statement holds for all of the zero-sum games corresponding to different iterations $i \in [n]$, then Algorithm 1 is a $\frac{1}{2}$-approximation of the optimum.

**Lemma 1.** *If* $\forall i \in [n]: \mathbf{E}\left[\mathcal{V}^{(i)}(\hat{z}_i, x^*_i)\right] \geq -\delta/n$ *for constant* $\delta > 0$*, then* $2\mathbf{E}\left[\mathcal{F}(\hat{\mathbf{z}})\right] \geq \mathcal{F}(\mathbf{x}^*) - \delta$.

**Algorithm 1:** (Vanilla) Continuous Randomized Bi-Greedy

**input**: function $\mathcal{F} : [0,1]^n \to [0,1]$ ;
**output**: vector $\hat{\mathbf{z}} = (\hat{z}_1, \ldots, \hat{z}_n) \in [0,1]^n$ ;
Initialize $\mathbf{X} \leftarrow (0, \ldots, 0)$ and $\mathbf{Y} \leftarrow (1, \ldots, 1)$ ;
**for** $i = 1$ *to* $n$ **do**

$\quad$ Find $Z_u, Z_l \in [0,1]$ such that $\begin{cases} Z_l \in \underset{z \in [0,1]}{\mathrm{argmax}} \, \mathcal{F}(z, \mathbf{Y}_{-i}) \\ Z_u \in \underset{z \in [0,1]}{\mathrm{argmax}} \, \mathcal{F}(z, \mathbf{X}_{-i}) \end{cases}$ ;

$\quad$ **if** $Z_u \leq Z_l$ **then**
$\quad\quad$ $\hat{z}_i \leftarrow Z_l$ ;
$\quad$ **else**

$\quad\quad$ $\forall z \in [Z_l, Z_u]$, let $\begin{cases} g(z) \triangleq \mathcal{F}(z, \mathbf{X}_{-i}) - \mathcal{F}(Z_l, \mathbf{X}_{-i}), \\ h(z) \triangleq \mathcal{F}(z, \mathbf{Y}_{-i}) - \mathcal{F}(Z_u, \mathbf{Y}_{-i}), \end{cases}$ ;

$\quad\quad$ Let $\alpha \triangleq g(Z_u)$ and $\beta \triangleq h(Z_l)$ ;  $\quad$ // note that $\alpha, \beta \geq 0$
$\quad\quad$ Let $\mathbf{r}(z) \triangleq (g(z), h(z))$ be a continuous two-dimensional curve in $[-1, \alpha] \times [-1, \beta]$ ;
$\quad\quad$ Compute `conc-env(r)` (i.e. positive-orthant concave envelope of $\mathbf{r}(t)$ as in Definition 3) ;
$\quad\quad$ Find point $\mathcal{P} \triangleq$ intersection of `conc-env(r)` and the line $h' - \beta = g' - \alpha$ on g-h plane ;
$\quad\quad$ Suppose $\mathcal{P} = \lambda \mathcal{P}_1 + (1 - \lambda)\mathcal{P}_2$, where $\lambda \in [0,1]$ and $\mathcal{P}_j = \mathbf{r}(z^{(j)}), z^{(j)} \in [Z_l, Z_u]$ for
$\quad\quad\quad\quad$ $j = 1, 2$, and both points are also on the `conc-env(r)` ;  $\quad$ // see Figure 2
$\quad\quad$ Randomly pick $\hat{z}_i$ such that $\begin{cases} \hat{z}_i \leftarrow z^{(1)} & \text{with probablity } \lambda \\ \hat{z}_i \leftarrow z^{(2)} & \text{o.w.} \end{cases}$ ;

$\quad$ Let $X_i \leftarrow \hat{z}_i$ and $Y_i \leftarrow \hat{z}_i$ ;  $\quad$ // after this, $\mathbf{X}$ and $\mathbf{Y}$ will agree at coordinate $i$

*Proof sketch.* Our bi-greedy approach, á la Buchbinder et al. [2012, 2015], revolves around analyzing the evolving values of three points: $\mathbf{X}^{(i)}$, $\mathbf{Y}^{(i)}$, and $\mathbf{O}^{(i)}$. These three points begin at all-zeroes, all-ones, and the optimum solution, respectively, and converge to the algorithm's final point. In each iteration, we aim to relate the total increase in value of the first two points with the decrease in value of the third point. If we can show that the former quantity is at least twice the latter quantity, then a telescoping sum proves that the algorithm's final choice of point scores at least half that of optimum.

The utility of our game is specifically engineered to compare the total increase in value of the first two points with the decrease in value of the third point. The positive term of the utility is half of this increase in value, and the negative term is a bound on how large in magnitude the decrease in value may be. As a result, an overall nonnegative utility implies that the increase beats the decrease by a factor of two, exactly the requirement for our bi-greedy approach to work. Finally, an additive slack of $\delta/n$ in the utility of each game sums over $n$ iterations for a total additive slack of $\delta$. $\quad\square$

***Analyzing the Zero-sum Games.*** Fix an iteration $i \in [n]$ of Algorithm 1. We then have the following.

**Proposition 1.** *If* ALG *plays the (randomized) strategy* $\hat{z}_i$ *as described in Algorithm 1, then we have* $\mathbf{E}\left[\mathcal{V}^{(i)}(\hat{z}_i, x_i^*)\right] \geq 0$ *against any strategy* $x_i^*$ *of* ADV.

*Proof of Proposition 1.* We do the proof by case analysis over two cases:

$\square$ **Case $Z_l \geq Z_u$ (*easy*):** See the supplementary materials for this case.

$\square$ **Case $Z_l < Z_u$ (*hard*):** In this case, ALG plays a mixed strategy over two points. To determine the two-point support, it considers the curve $\mathbf{r} = \{(g(z), h(z))\}_{z \in [Z_l, Z_u]}$ and finds a point $\mathcal{P}$ on `conc-env(r)` (i.e., Definition 3) that lies on the line $h' - \beta = g' - \alpha$, where recall that $\alpha = g(Z_u) \geq 0$ and $\beta = g(Z_l) \geq 0$ (as $Z_u$ and $Z_l$ are the maximizers of $\mathcal{F}(z, \mathbf{X}_{-i}^{(i-1)})$ and $\mathcal{F}(z, \mathbf{Y}_{-i}^{(i-1)})$ respectively). Because this point is on the concave envelope it should be a convex combination of two points on the curve $\mathbf{r}(z)$. Lets say $\mathcal{P} = \lambda \mathcal{P}_1 + (1 - \lambda)\mathcal{P}_2$, where $\mathcal{P}_1 = \mathbf{r}(z^{(1)})$ and $\mathcal{P}_2 = \mathbf{r}(z^{(2)})$, and $\lambda \in [0,1]$. The final strategy of ALG is a mixed strategy over $\{z^{(1)}, z^{(2)}\}$ with probabilities $(\lambda, 1 - \lambda)$. Fixing any mixed strategy of ALG over two points $\mathcal{P}_1 = (g_1, h_1)$ and $\mathcal{P}_2 = (g_2, h_2)$ with probabilities

$(\lambda, 1 - \lambda)$ (denoted by $F_{\mathcal{P}}$), define the ADV's *positive region*, i.e.

$$(g', h') \in [-1, 1] \times [-1, 1] : \quad \mathbf{E}_{(g,h) \sim F_{\mathcal{P}}} \left[ \frac{1}{2}g + \frac{1}{2}h - \max(g' - g, h' - h) \right] \geq 0.$$

Now, suppose ALG plays a mixed strategy with the property that its corresponding ADV's positive region covers the entire curve $\{g(z), h(z)\}_{z \in [0,1]}$. Then, for any strategy $x_i^*$ of ADV the expected utility of ALG is non-negative. In the rest of the proof, we geometrically characterize the ADV's positive region against a mixed strategy of ALG over any 2-point support, and then we show for the particular choice of $\mathcal{P}_1, \mathcal{P}_2$ and $\lambda$ in Algorithm 1 the positive region covers the entire curve $\{g(z), h(z)\}_{z \in [0,1]}$.

**Lemma 2.** *Suppose ALG plays a 2-point mixed strategy over $\mathcal{P}_1 = \mathbf{r}(z^{(1)}) = (g_1, h_1)$ and $\mathcal{P}_2 = \mathbf{r}(z^{(1)}) = (g_2, h_2)$ with probabilities $(\lambda, 1 - \lambda)$, and w.l.o.g. $h_1 - g_1 \geq h_2 - g_2$. Then ADV's positive region is the pentagon $(\mathcal{M}_0, \mathcal{M}_1, \mathcal{Q}_1, \mathcal{Q}_2, \mathcal{M}_2)$, where $\mathcal{M}_0 = (-1, -1)$ and (see Figure 2):*

1. $\mathcal{M}_1 = \left( -1, \lambda(\frac{3}{2}h_1 + \frac{1}{2}g_1) + (1 - \lambda)(\frac{3}{2}h_2 + \frac{1}{2}g_2) \right)$,

2. $\mathcal{M}_2 = \left( \lambda(\frac{3}{2}g_1 + \frac{1}{2}h_1) + (1 - \lambda)(\frac{3}{2}g_2 + \frac{1}{2}h_2), -1 \right)$,

3. $\mathcal{Q}_1$ *is the intersection of the lines leaving $\mathcal{P}_1$ with slope 1 and leaving $\mathcal{M}_1$ along the g-axis,*

4. $\mathcal{Q}_2$ *is the intersection of the lines leaving $\mathcal{P}_2$ with slope 1 and leaving $\mathcal{M}_2$ along the h-axis.*

By applying Lemma 2, we have the following main technical lemma. The proof is geometric and is pictorially visible in Figure 2. This lemma finishes the proof of Proposition 1.

**Lemma 3** (***main lemma***). *If ALG plays the two point mixed strategy described in Algorithm 1, then for every $x_i^* \in [0, 1]$ the point $(g', h') = (g(x_i^*), h(x_i^*))$ is in the ADV's positive region.*

*Proof sketch.* For simplicity assume $Z_l = 0$ and $Z_u = 1$. To understand the ADV's positive region that results from playing a two-point mixed strategy by ALG, we consider the positive region that results from playing a one point pure strategy. When ALG chooses a point $(g, h)$, the positive term of the utility is one-half of its one-norm. The negative term of the utility is the worse between how much the ADV's point is above ALG's point, and how much it is to the right of ALG's point. The resulting positive region is defined by an upper boundary $g' \leq \frac{3}{2}g + \frac{1}{2}h$ and a right boundary $h' \leq \frac{1}{2}g + \frac{3}{2}h$.

Next, let's consider what happens when we pick point $(g_1, h_1)$ with probability $\lambda$ and point $(g_2, h_2)$ with probability $(1 - \lambda)$. We can compute the expected point: let $(g_3, h_3) = \lambda(g_1, h_1) + (1 - \lambda)(g_2, h_2)$. As suggested by Lemma 2, the positive region for our mixed strategy has three boundary conditions: an upper boundary, a right boundary, and a corner-cutting boundary. The first two boundary conditions correspond to a pure strategy which picks $(g_3, h_3)$. By design, $(g_3, h_3)$ is located so that these boundaries cover the entire $[-1, \alpha] \times [-1, \beta]$ rectangle. This leaves us with analyzing the corner-cutting boundary, which is the focus of Figure 2. As it turns out, the intersections of this boundary with the two other boundaries lie on lines of slope 1 extending from $(g_j, h_j)_{j=1,2}$. If we consider the region between these two lines, the portion under the envelope (where the curve $\mathbf{r}$ may lie) is distinct from the portion outside the corner-cutting boundary. However, if $\mathbf{r}$ were to ever violate the corner-cutting boundary condition without violating the other two boundary conditions, it must do so in this region. Hence the resulting positive region covers the entire curve $\mathbf{r}$, as desired. □

## 2.2 Polynomial-time Implementation under Lipschitz Continuity: Overview

At each iteration, Algorithm 1 interfaces with $\mathcal{F}$ in two ways: (i) when performing optimization to compute $Z_l, Z_u$ and (ii) when computing the upper-concave envelope. In both cases, we are concerned with univariate projections of $\mathcal{F}$, namely $\mathcal{F}(z, \mathbf{X}_{-i})$ and $\mathcal{F}(z, \mathbf{Y}_{-i})$. Assuming $\mathcal{F}$ is coordinate-wise Lipschitz continuous with constant $C > 0$, we choose a small $\epsilon > 0$ and take periodic samples at $\epsilon$-spaced intervals from each one of these functions, for a total of $O(\frac{1}{\epsilon})$ samples.

To perform task (i), we simply return the the sample which resulted in the maximum function value. Since the actual maximum is $\epsilon$-close to one of the samples, our maximum is at most an additive $\epsilon C$ lower in value. To perform task (ii), we use these samples to form an approximate $\mathbf{r}(z)$ curve, denoted by $\hat{\mathbf{r}}(z)$. Note that we then proceed exactly as described in Algorithm 1 to pick a (randomized) strategy

$\hat{z}_i$ using $\hat{\mathbf{r}}(z)$. Note that ADV can actually choose a point on the exact curve $\mathbf{r}(z)$. However the point she chooses is close to one of our samples and hence is at most an additive $\epsilon C$ better in value with respect to functions $g(.)$ and $h(.)$. Furthermore, we can compute the upper-concave envelope $\hat{\mathbf{r}}(z)$ in time linear in the number of samples using Graham's algorithm [Graham, 1972]. Roughly speaking, this is because we can go through the samples in order of $z$-coordinate, avoiding the sorting cost of running Graham's on completely unstructured data. Formally, we have the following proposition. See the supplementary materials for detailed implementations (Algorithm 3 and Algorithm 4).

**Proposition 2.** *If $\mathcal{F}$ is coordinate-wise Lipschitz continuous with constant $C > 0$, then Algorithm 1 can be implemented with $O(n^2/\epsilon)$ calls to $\mathcal{F}$ and returning a (randomized) point $\hat{\mathbf{z}}$ s.t.*

$$2\mathbf{E}\left[\mathcal{F}(\hat{\mathbf{z}})\right] \geq \mathcal{F}(\mathbf{x}^*) - 2C\epsilon, \qquad \text{where } \mathbf{x}^* \in \underset{\boldsymbol{x}\in[0,1]^n}{\operatorname{argmax}} \mathcal{F}(\boldsymbol{x}) \text{ is the optimal solution.}$$

# 3 Strong DR-SM Maximization: Binary-Search Bi-Greedy

Our second result is a fast binary search algorithm, achieving the tight $\frac{1}{2}$-approximation factor (up to additive error $\delta$) in quasi-linear time in $n$, but only for the special case of strong DR-SM functions (a.k.a. DR-submodular); see Definition 1. This algorithm leverages the coordinate-wise concavity to identify a coordinate-wise *monotone equilibrium condition*. In each iteration, it hunts for an equilibrium point by using binary search. Satisfying the equilibrium at each iteration then guarantees the desired approximation factor. Formally we propose Algorithm 2. As a technical assumption, we

---

**Algorithm 2:** Binary-Search Continuous Bi-greedy

**input**: function $\mathcal{F} : [0,1]^n \to [0,1]$, error $\epsilon > 0$ ;
**output**: vector $\hat{\mathbf{z}} = (\hat{z}_1, \ldots, \hat{z}_n) \in [0,1]^n$ ;
Initialize $\mathbf{X} \leftarrow (0, \ldots, 0)$ and $\mathbf{Y} \leftarrow (1, \ldots, 1)$ ;
**for** $i = 1$ *to* $n$ **do**
    **if** $\frac{\partial \mathcal{F}}{\partial x_i}(0, \mathbf{X}_{-i}) \leq 0$ **then**
        $\hat{z}_i \leftarrow 0$
    **else if** $\frac{\partial \mathcal{F}}{\partial x_i}(1, \mathbf{Y}_{-i}) \geq 0$ **then**
        $\hat{z}_i \leftarrow 1$
    **else**
        // we do binary search.
        **while** $Y_i - X_i > \epsilon/n$ **do**
            Let $\hat{z}_i \leftarrow \frac{X_i + Y_i}{2}$ ;
            **if** $\frac{\partial \mathcal{F}}{\partial x_i}(\hat{z}_i, \mathbf{X}_{-i}) \cdot (1 - \hat{z}_i) + \frac{\partial \mathcal{F}}{\partial x_i}(\hat{z}_i, \mathbf{Y}_{-i}) \cdot \hat{z}_i > 0$ **then**
                // we need to increase $w_i$.
                Set $X_i \leftarrow \hat{z}_i$ ;
            **else**
                // we need to decrease $w_i$.
                Set $Y_i \leftarrow \hat{z}_i$ ;
    Let $X_i \leftarrow \hat{z}_i$ and $Y_i \leftarrow \hat{z}_i$ ;    // after this, $\mathbf{X}$ and $\mathbf{Y}$ will agree at coordinate $i$

---

assume $\mathcal{F}$ is Lipschitz continuous with some constant $C > 0$, so that we can relate the precision of our binary search with additive error. We arrive at the theorem, whose proof is in the supplement.

**Theorem 2.** *If $\mathcal{F}(.)$ is non-negative and DR-submodular (a.k.a Strong DR-SM) and is coordinate-wise Lipschitz continuous with constant $C > 0$, then Algorithm 2 runs in time $O\left(n\log\left(\frac{n}{\epsilon}\right)\right)$ and is a deterministic $\frac{1}{2}$-approximation algorithm up to $O(\epsilon)$ additive error, i.e. returns $\hat{\mathbf{z}} \in [0,1]^n$ s.t.*

$$2\mathcal{F}(\hat{\mathbf{z}}) \geq \mathcal{F}(\mathbf{x}^*) - 2C\epsilon, \qquad \text{where } \mathbf{x}^* \in \underset{\boldsymbol{x}\in[0,1]^n}{\operatorname{argmax}} \mathcal{F}(\boldsymbol{x}) \text{ is the optimal solution.}$$

**Running Time.** If we show that $f(z) \triangleq \frac{\partial \mathcal{F}}{\partial x_i}(z, \mathbf{X}_{-i})(1 - z) + \frac{\partial \mathcal{F}}{\partial x_i}(z, \mathbf{Y}_{-i})z$ is monotone non-increasing in $z$, then clearly the binary search terminates in $O\left(\log(n/\epsilon)\right)$ steps (note that the algorithm only does binary search in the case when $f(0) > 0$ and $f(1) < 0$). To see the monotonicity,

$$f'(z) = (1-z)\frac{\partial^2 \mathcal{F}}{\partial x_i^2}(z, \mathbf{X}_{-i}) + z\frac{\partial^2 \mathcal{F}}{\partial x_i^2}(z, \mathbf{Y}_{-i}) + \left(\frac{\partial \mathcal{F}}{\partial x_i}(z, \mathbf{Y}_{-i}) - \frac{\partial \mathcal{F}}{\partial x_i}(z, \mathbf{X}_{-i})\right) \leq 0$$

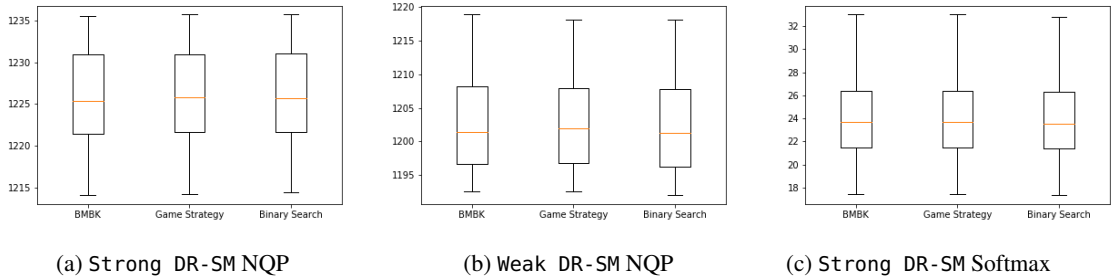

(a) Strong DR-SM NQP　　　　(b) Weak DR-SM NQP　　　　(c) Strong DR-SM Softmax

Figure 3: Box and whisker plots of our experimental results.

where the inequality holds due to strong DR-SM and the fact that all of the Hessian entries (including diagonal) are non-positive. Hence the total running time is $O\left(n\log(n/\epsilon)\right)$.

## 4　Experimental Results

We empirically measure the solution quality of three algorithms: Algorithm 1 (GAME), Algorithm 2 (BINARY) and the Bi-Greedy algorithm of Bian et al. [2017b] (BMBK). These are all based on a double-greedy framework, which we implemented to iterate over coordinates in a random order. These algorithms also do not solely rely on oracle access to the function; they invoke one-dimensional optimizers, concave envelopes, and derivatives. We implement the first and the second (Algorithm 3 and Algorithm 4 in the supplement), and numerically compute derivatives by discretization. We consider two application domains, namely Non-concave Quadratic Programming (NQP) [Bian et al., 2017b, Kim and Kojima, 2003, Luo et al., 2010], under both strong-DR and weak-DR, and maximization of softmax extension for MAP inference of determinantal point process[Kulesza et al., 2012, Gillenwater et al., 2012]. Each experiment consists of twenty repeated trials. For each experiment, we use $n = 100$ dimensional functions. Our experiments were implemented in python. See the supplementary materials for the detailed specifics of each experiment. The results of our experiments are in Table 1, and the corresponding box and whisker plots are in Figure 3. The data suggest that for all three experiments the three algorithms obtain very similar objective values. For example, in the weak-DR NQP experiment, all three algorithms have standard deviations around 6 while their means differ by less than 1.

| | NQP, $\forall i, j : H_{i,j} \leq 0$, (strong-DR) | NQP, $\forall i \neq j : H_{i,j} \leq 0$, (weak-DR) | Softmax Ext. (strong-DR) |
|---|---|---|---|
| GAME | $1225.416454 \pm 8.201871$ | $1200.860403 \pm 6.009484$ | $24.056934 \pm 3.794209$ |
| BINARY | $1225.392136 \pm 8.203827$ | $1200.248876 \pm 6.088293$ | $23.945428 \pm 3.770932$ |
| BMBK | $1225.339063 \pm 8.141104$ | $1200.798114 \pm 5.975035$ | $24.055435 \pm 3.796350$ |

Table 1: Experimental results listing mean and standard deviation over $T = 20$ repeated trials with dimension $n = 100$.

## 5　Conclusion

We proposed a tight approximation algorithm for continuous submodular maximization, and a qausi-linear time tight approximation algorithm for the special case of DR-submodular maxmization. Our experiments also verify the applicability of these algorithms in practical domains in machine learning. One interesting avenue for future research is to generalize our techniques to the maximization over any arbitrary separable convex set, which will result in a broader application domain.

## Footnotes

[3]Our results also extend easily to arbitrary axis-aligned boxes (i.e., "box constraints").

[4]More generally, the function only has to be nonnegative at the points $\vec{0}$ and $\vec{1}$.

[5]See the supplement for more details on these applications.

[6]An instance of the latter problem can be converted to one of the former by extending the given set function $f$ (with domain viewed as $\{0,1\}^n$) to its multilinear extension $\mathcal{F}$ defined on the hypercube (where $\mathcal{F}(\mathbf{x}) = \sum_{S \subseteq [n]} \prod_{i \in S} x_i \prod_{i \notin S}(1 - x_i)f(S)$). Sampling based on an $\alpha$-approximate solution for the multilinear extension yields an equally good approximate solution to the original problem.

[7]However, after regularization the function still remains submodular, but can lose coordinate-wise concavity.

[8]Such an assumption is necessary, since otherwise the single-dimensional problem amounts to optimizing an arbitrary function and is hence intractable. Prior works, e.g,. Bian et al. [2017b] and Bian et al. [2017a], implicitly require such an assumption to perform single-dimensional optimization.

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
