[Supplementary Material]

## Supplementary Materials

## Equivalent definitions of weakly and strongly DR-SM functions.

**Proposition 3** ([Bian et al., 2017b]). *Suppose $\mathcal{F} : [0,1]^n \to [0,1]$ is continuous and twice differentiable, and $\mathbf{H}$ is the Hessian of $\mathcal{F}$, i.e. $\forall i, j \in [n]$, $H_{ij} \triangleq \frac{\partial^2 \mathcal{F}}{\partial x_i \partial x_j}$. The followings are equivalent:*

1. *$\mathcal{F}$ satisfies the* `weak` `DR-SM` *property as in Definition 1.*

2. *Continuous submodularity: $\forall \boldsymbol{x}, \boldsymbol{y} \in [0,1]^n$, $\mathcal{F}(\boldsymbol{x}) + \mathcal{F}(\boldsymbol{y}) \geq \mathcal{F}(\boldsymbol{x} \vee \boldsymbol{y}) + \mathcal{F}(\boldsymbol{x} \wedge \boldsymbol{y})$.*

3. *$\forall i \neq j \in [n]$, $H_{ij} \leq 0$, i.e., all off-diagonal entries of Hessian are non-positive.*

*Also, the following statements are equivalent:*

1. *$\mathcal{F}$ satisfies the* `strong` `DR-SM` *property as in Definition 1.*

2. *$\mathcal{F}(.)$ is coordinate-wise concave along all the coordinates and is continuous submodular, i.e. $\forall \boldsymbol{x}, \boldsymbol{y} \in [0,1]^n$, $\mathcal{F}(\boldsymbol{x}) + \mathcal{F}(\boldsymbol{y}) \geq \mathcal{F}(\boldsymbol{x} \vee \boldsymbol{y}) + \mathcal{F}(\boldsymbol{x} \wedge \boldsymbol{y})$*

3. *$\forall i, j \in [n]$, $H_{ij} \leq 0$, i.e., all entries of Hessian are non-positive.*

## Missing proofs of Section 2.

***Proof of Lemma 1.*** Consider a realization of $\hat{z}_i$ where $\hat{z}_i \geq x_i^*$. We have:

$$
\begin{aligned}
\mathcal{F}(\mathbf{O}^{(i-1)}) - \mathcal{F}(\mathbf{O}^{(i)}) &= \mathcal{F}(\hat{z}_1, \ldots, \hat{z}_{i-1}, x_i^*, x_{i+1}^*, \ldots, x_n^*) - \mathcal{F}(\hat{z}_1, \ldots, \hat{z}_{i-1}, \hat{z}_i, x_{i+1}^*, \ldots, x_n^*) \\
&\leq \mathcal{F}(\hat{z}_1, \ldots, \hat{z}_{i-1}, x_i^*, 1, \ldots, 1) - \mathcal{F}(\hat{z}_1, \ldots, \hat{z}_{i-1}, \hat{z}_i, 1, \ldots, 1) \\
&= \left( \mathcal{F}(x_i^*, \mathbf{Y}_{-i}^{(i-1)}) - \mathcal{F}(Z_u, \mathbf{Y}_{-i}^{(i-1)}) \right) - \left( \mathcal{F}(\hat{z}_i, \mathbf{Y}_{-i}^{(i-1)}) - \mathcal{F}(Z_u, \mathbf{Y}_{-i}^{(i-1)}) \right) \\
&= h(x_i^*) - h(\hat{z}_i), \quad\quad (2)
\end{aligned}
$$

where the inequality holds due to `weak` `DR-SM`. Similarly, for a a realization of $\hat{z}_i$ where $\hat{z}_i \leq x_i^*$:

$$
\begin{aligned}
\mathcal{F}(\mathbf{O}^{(i-1)}) - \mathcal{F}(\mathbf{O}^{(i)}) &= \mathcal{F}(\hat{z}_1, \ldots, \hat{z}_{i-1}, x_i^*, x_{i+1}^*, \ldots, x_n^*) - \mathcal{F}(\hat{z}_1, \ldots, \hat{z}_{i-1}, \hat{z}_i, x_{i+1}^*, \ldots, x_n^*) \\
&\leq \mathcal{F}(\hat{z}_1, \ldots, \hat{z}_{i-1}, x_i^*, 0, \ldots, 0) - \mathcal{F}(\hat{z}_1, \ldots, \hat{z}_{i-1}, \hat{z}_i, 0, \ldots, 0) \\
&= \left( \mathcal{F}(x_i^*, \mathbf{X}_{-i}^{(i-1)}) - \mathcal{F}(Z_l, \mathbf{X}_{-i}^{(i-1)}) \right) - \left( \mathcal{F}(\hat{z}_i, \mathbf{X}_{-i}^{(i-1)}) - \mathcal{F}(Z_l, \mathbf{X}_{-i}^{(i-1)}) \right) \\
&= g(x_i^*) - g(\hat{z}_i) \quad\quad (3)
\end{aligned}
$$

Putting eq. (2) and eq. (3) together, for every realization $\hat{z}_i$ we have:

$$
F(\mathbf{O}^{(i-1)}) - \mathcal{F}(\mathbf{O}^{(i)}) \leq \max \left( g(x_i^*) - g(\hat{z}_i), h(x_i^*) - h(\hat{z}_i) \right) \quad\quad (4)
$$

Moreover, consider the term $\mathcal{F}(\mathbf{X}^{(i)}) - \mathcal{F}(\mathbf{X}^{(i-1)})$. We have:

$$
\begin{aligned}
\mathcal{F}(\mathbf{X}^i) - \mathcal{F}(\mathbf{X}^{(i-1)}) &= \mathcal{F}(\hat{z}_1, \ldots, \hat{z}_{i-1}, \hat{z}_i, 0, \ldots, 0) - \mathcal{F}(\hat{z}_1, \ldots, \hat{z}_{i-1}, 0, 0, \ldots, 0) \\
&= g(\hat{z}_i) - g(0) = g(\hat{z}_i) + \mathcal{F}(Z_l, \mathbf{X}_{-i}^{(i-1)}) - \mathcal{F}(\mathbf{X}^{(i-1)}) \\
&\geq g(\hat{z}_i) + \mathcal{F}(Z_l, \mathbf{Y}_{-i}^{(i-1)}) - \mathcal{F}(0, \mathbf{Y}_{-i}^{(i-1)}) \geq g(\hat{z}_i) \quad\quad (5)
\end{aligned}
$$

where the first inequality holds due to `weak` `DR-SM` property and the second inequity holds as $Z_l \in \underset{z \in [0,1]}{\operatorname{argmax}} \mathcal{F}(z, \mathbf{Y}_{-i}^{(i-1)})$. Similarly, consider the term $\mathcal{F}(\mathbf{Y}^{(i)}) - \mathcal{F}(\mathbf{Y}^{(i-1)})$. We have:

$$
\begin{aligned}
\mathcal{F}(\mathbf{Y}^{(i)}) - \mathcal{F}(\mathbf{Y}^{(i-1)}) &= \mathcal{F}(\hat{z}_1, \ldots, \hat{z}_{i-1}, \hat{z}_i, 1, \ldots, 1) - \mathcal{F}(\hat{z}_1, \ldots, \hat{z}_{i-1}, 1, 1, \ldots, 1) \\
&= h(\hat{z}_i) - h(1) = h(\hat{z}_i) + \mathcal{F}(Z_u, \mathbf{Y}_{-i}^{(i-1)}) - \mathcal{F}(\mathbf{Y}^{(i-1)}) \\
&\geq h(\hat{z}_i) + \mathcal{F}(Z_u, \mathbf{X}_{-i}^{(i-1)}) - \mathcal{F}(1, \mathbf{X}_{-i}^{(i-1)}) \geq h(\hat{z}_i) \quad\quad (6)
\end{aligned}
$$

where the first inequality holds due to weak DR-SM and the second inequity holds as $Z_u \in \underset{z \in [0,1]}{\mathrm{argmax}}\, \mathcal{F}(z, \mathbf{X}_{-i}^{(i-1)})$. By eq. (4), eq. (5), eq. (6), and the fact that $\mathcal{F}(\mathbf{0}) + \mathcal{F}(\mathbf{1}) \geq 0$, we have:

$$0 \leq \sum_{i=1}^{n} \mathbf{E}\left[\mathcal{V}^{(i)}(\hat{z}_i, x_i^*)\right] = \sum_{i=1}^{n} \left(\frac{1}{2}\mathbf{E}\left[g(\hat{z}_i)\right] + \frac{1}{2}\mathbf{E}\left[h(\hat{z}_i)\right] - \mathbf{E}\left[\max\left(g(x_i^*) - g(\hat{z}_i), h(x_i^*) - h(\hat{z}_i)\right)\right]\right)$$

$$\leq \frac{1}{2}\mathbf{E}\left[\sum_{i=1}^{n}\left(\mathcal{F}(\mathbf{X}^{(i)}) - \mathcal{F}(\mathbf{X}^{(i-1)})\right) + \sum_{i=1}^{n}\left(\mathcal{F}(\mathbf{Y}^{(i)}) - \mathcal{F}(\mathbf{Y}^{(i-1)})\right) - 2\sum_{i=1}^{n}\left(\mathcal{F}(\mathbf{O}^{(i-1)}) - \mathcal{F}(\mathbf{O}^{(i)})\right)\right]$$

$$= \mathbf{E}\left[\frac{\mathcal{F}(\mathbf{X}^{(n)}) - \mathcal{F}(\mathbf{X}^{(0)})}{2} + \frac{\mathcal{F}(\mathbf{Y}^{(n)}) - \mathcal{F}(\mathbf{Y}^{(0)})}{2} - \mathcal{F}(\mathbf{O}^{(0)}) + \mathcal{F}(\mathbf{O}^{(n)})\right]$$

$$\leq \mathbf{E}\left[\frac{\mathcal{F}(\hat{\mathbf{z}})}{2} + \frac{\mathcal{F}(\hat{\mathbf{z}})}{2} - \mathcal{F}(\mathbf{x}^*) + \mathcal{F}(\hat{\mathbf{z}})\right] = 2\mathbf{E}\left[\mathcal{F}(\hat{\mathbf{z}})\right] - \mathcal{F}(\mathbf{x}^*) \qquad \square$$

***Proof of Lemma 2.*** We start by a technical lemma, showing a single-crossing property of the g-h curve of a weak DR submodular function $\mathcal{F}(.)$, and we then characterize the region using this lemma.

**Lemma 4.** *The univariate function $d(z) = h(z) - g(z)$ is monotone non-increasing.*

*Proof.* By using weak DR-SM property of $\mathcal{F}(.)$ the proof is immediate, as for any $\delta \geq 0$,

$$d(z+\delta) - d(z) = \left(\mathcal{F}(z+\delta, \mathbf{Y}_{-i}^{(i-1)}) - \mathcal{F}(z, \mathbf{Y}_{-i}^{(i-1)})\right) - (F(z+\delta, \mathbf{X}_{-i}^{(i-1)}) - F(z+\delta, \mathbf{X}_{-i}^{(i-1)})) \leq 0,$$

where the inequality holds due to the fact that $\mathbf{Y}_{-i}^{(i-1)} \geq \mathbf{X}_{-i}^{(i-1)}$ and $\delta \geq 0$. $\qquad \square$

Being equipped with Lemma 4, the positive region is the set of all points $(g', h') \in [-1, 1]^2$ such that

$$\mathbf{E}_{(g,h)\sim F_{\mathcal{P}}}\left[\frac{1}{2}g + \frac{1}{2}h - \max(g'-g, h'-h)\right]$$

$$= \lambda\left(\frac{1}{2}g_1 + \frac{1}{2}h_1 - \max(g'-g_1, h'-h_1)\right) + (1-\lambda)\left(\frac{1}{2}g_2 + \frac{1}{2}h_2 - \max(g'-g_2, h'-h_2)\right) \geq 0$$

The above inequality defines a polytope. Our goal is to find the vertices and faces of this polytope. Now, to this end, we only need to consider three cases: 1) $h' - g' \geq h_1 - g_1$, 2) $h_2 - g_2 \leq h' - g' \leq h_1 - g_1$ and 3) $h' - g' \leq h_2 - g_2$ (note that $h_1 - g_1 \geq h_2 - g_2$). From the first and third case we get the half-spaces $h' \leq \lambda(\frac{3}{2}h_1 + \frac{1}{2}g_1) + (1-\lambda)(\frac{3}{2}h_2 + \frac{1}{2}g_2)$ and $g' \leq \lambda(\frac{3}{2}g_1 + \frac{1}{2}h_1) + (1-\lambda)(\frac{3}{2}g_2 + \frac{1}{2}h_2)$ respectively, that form two of the faces of the positive-region polytope. From the second case, we get another half-space, but the observation is that the transition from first case to second case happens when $h' - g' = h_1 - g_1$, i.e. on a line with slope one leaving $\mathcal{P}_1$, and transition from second case to the third case happens when $h' - g' = h_2 - g_2$, i.e. on a line with slope one leaving $\mathcal{P}_2$. Therefore, the second half-space is the region under the line connecting two points $\mathcal{Q}_1$ and $\mathcal{Q}_2$, where $\mathcal{Q}_1$ is the intersection of $h' = \lambda(\frac{3}{2}h_1 + \frac{1}{2}g_1) + (1-\lambda)(\frac{3}{2}h_2 + \frac{1}{2}g_2)$ and the line leaving $\mathcal{P}_1$ with slope one (point $\mathcal{Q}_1$), and $\mathcal{Q}_2$ is the intersection of $g' = \lambda(\frac{3}{2}g_1 + \frac{1}{2}h_1) + (1-\lambda)(\frac{3}{2}g_2 + \frac{1}{2}h_2)$ and the line leaving $\mathcal{P}_2$ with slope one (point $\mathcal{Q}_2$). The line segment $\mathcal{Q}_1 - \mathcal{Q}_2$ defines another face of the positive region polytope, and $\mathcal{Q}_1$ and $\mathcal{Q}_2$ will be two vertices on this face. By intersecting the three mentioned half-spaces with $g' \geq -1$ and $h \geq -1$ (which define the two remaining faces of the positive region polytope), the postive region will be the polytope defined by the pentagon $(\mathcal{M}_0, \mathcal{M}_1, \mathcal{Q}_1, \mathcal{Q}_2, \mathcal{M}_2)$, as claimed (see Figure 2 for a pictorial proof). $\qquad \square$

***Proof of the "Case $\mathbf{Z}_l \geq \mathbf{Z}_u$ (easy)" of Proposition 1***. In this case, the algorithm plays a deterministic strategy $\hat{z}_i = Z_l$. We therefore have:

$$\mathcal{V}^{(i)}(\hat{z}_i, x_i^*) = \frac{1}{2}g(\hat{z}_i) + \frac{1}{2}h(\hat{z}_i) - \max\left(g(x_i^*) - g(\hat{z}_i), h(x_i^*) - h(\hat{z}_i)\right) \geq \min(g(\hat{z}_i) - g(x_i^*), 0)$$

where the inequality holds because $g(\hat{z}_i) = g(Z_l) = 0$, and also $Z_l \in \underset{z \in [0,1]}{\mathrm{argmax}}\, \mathcal{F}(z, \mathbf{Y}_{-l}^{(i)})$ and so:

- $h(\hat{z}_i) = h(Z_l) = \mathcal{F}(Z_l, \mathbf{Y}_{-i}^{(i)}) - \mathcal{F}(Z_u, \mathbf{Y}_{-i}^{(i)}) \geq 0$

- $h(x_i^*) - h(\hat{z}_i) = \mathcal{F}(x_i^*, \mathbf{Y}_{-i}^{(i-1)}) - \mathcal{F}(Z_l, \mathbf{Y}_{-i}^{(i-1)}) \leq 0$

To complete the proof for this case, it is only remained to show $g(\hat{z}_i) - g(x_i^*) \geq 0$. As $Z_l \geq Z_u$, for any given $x_i^* \in [0,1]$ either $x_i^* \leq Z_l$ or $x_i^* \geq Z_u$ (or both). If $x_i^* \leq Z_l$ then:

$$g(\hat{z}_i) - g(x_i^*) = -g(x_i^*) = \mathcal{F}(Z_l, \mathbf{X}_{-i}^{(i-1)}) - \mathcal{F}(x_i^*, \mathbf{X}_{-i}^{(i-1)}) \geq \mathcal{F}(Z_l, \mathbf{Y}_{-i}^{(i-1)}) - \mathcal{F}(x_i^*, \mathbf{Y}_{-i}^{(i-1)}) \geq 0$$

where the first inequality uses weak DR-SM property and the second inequality uses the fact $Z_l \in \operatorname*{argmax}_{z \in [0,1]} \mathcal{F}(z, \mathbf{Y}_{-i}^{(i)})$. If $x_i^* \leq Z_u$, we then have:

$$
\begin{aligned}
g(\hat{z}_i) - g(x_i^*) &= \mathcal{F}(Z_l, \mathbf{X}_{-i}^{(i-1)}) - \mathcal{F}(x_i^*, \mathbf{X}_{-i}^{(i-1)}) \\
&= \mathcal{F}(Z_l, \mathbf{X}_{-i}^{(i-1)}) - \mathcal{F}(Z_u, \mathbf{X}_{-i}^{(i-1)}) + \mathcal{F}(Z_u, \mathbf{X}_{-i}^{(i-1)}) - \mathcal{F}(x_i^*, \mathbf{X}_{-i}^{(i-1)}) \\
&\geq \left(\mathcal{F}(Z_l, \mathbf{Y}_{-i}^{(i-1)}) - \mathcal{F}(Z_u, \mathbf{Y}_{-i}^{(i-1)})\right) + \left(\mathcal{F}(Z_u, \mathbf{X}_{-i}^{(i-1)}) - \mathcal{F}(x_i^*, \mathbf{X}_{-i}^{(i-1)})\right) \geq 0
\end{aligned}
$$

where the first inequality uses weak DR-SM property and the second inequality holds because both terms are non-negative, following the fact that:

$$Z_l \in \operatorname*{argmax}_{z \in [0,1]} \mathcal{F}(z, \mathbf{Y}_{-i}^{(i)}) \qquad \text{and} \qquad Z_u \in \operatorname*{argmax}_{z \in [0,1]} \mathcal{F}(z, \mathbf{X}_{-i}^{(i)})$$

Therefore, we finish the proof of the easy case. $\qquad\square$

***Proof of the main technical lemma, i.e., Lemma 3.*** First of a all, we claim any ADV's strategy $x_i^* \in [0, Z_l)$ (or $x_i^* \in (Z_u, 1]$) is weakly dominated by $Z_l$ (or $Z_u$) if ALG plays a (randomized) strategy $\hat{z}_i \in [Z_l, Z_u]$. To see this, if $x_i^* \in [0, Z_l)$,

$$
\begin{aligned}
&\max\left(g(x_i^*) - g(\hat{z}_i), h(x_i^*) - h(\hat{z}_i)\right) \\
&= \max\left(\mathcal{F}(x_i^*, \mathbf{X}_{-i}^{(i-1)}) - \mathcal{F}(\hat{z}_i, \mathbf{X}_{-i}^{(i-1)}), \mathcal{F}(x_i^*, \mathbf{Y}_{-i}^{(i-1)}) - \mathcal{F}(\hat{z}_i, \mathbf{Y}_{-i}^{(i-1)})\right) \\
&= \mathcal{F}(x_i^*, \mathbf{Y}_{-i}^{(i-1)}) - \mathcal{F}(\hat{z}_i, \mathbf{Y}_{-i}^{(i-1)}) \leq \mathcal{F}(Z_l, \mathbf{Y}_{-i}^{(i-1)}) - \mathcal{F}(\hat{z}_i, \mathbf{Y}_{-i}^{(i-1)}) \\
&= h(Z_l) - h(\hat{z}_i) \leq \max\left(g(Z_l) - g(\hat{z}_i), h(Z_l) - h(\hat{z}_i)\right)
\end{aligned}
$$

and therefore $\mathcal{V}^{(i)}(\hat{z}_i, Z_l) \leq \mathcal{V}^{(i)}(\hat{z}_i, x_i^*)$ for any $x_i^* \in [0, Z_l)$. Similarly, $\mathcal{V}^{(i)}(\hat{z}_i, Z_u) \leq \mathcal{V}^{(i)}(\hat{z}_i, x_i^*)$ for any $x_i^* \in (Z_u, 1]$. So, without loss of generality, we can assume ADV's strategy $x_i^*$ is in $[Z_l, Z_u]$.

Now, consider the curve $\mathbf{r} = \{(g(z), h(z)\}_{z \in [Z_l, Z_u]}$ as in Figure 2. ALG's strategy is a 2-point mixed strategy over $\mathcal{P}_1 = (g_1, h_1) = \mathbf{r}(z^{(1)})$ and $\mathcal{P}_2 = (g_2, h_2) = \mathbf{r}(z^{(1)})$, where these two points are on different sides of the line $\mathcal{L} : h' - \beta = g' - \alpha$ (or both of them are on the line $\mathcal{L}$). Without loss of generality, assume $h_1 - g_1 \geq \beta - \alpha \geq h_2 - g_2$. Note that $\mathbf{r}(Z_l) = (0, \beta)$ is above the line $\mathcal{L}$ and $\mathbf{r}(Z_l) = (\alpha, 0)$ is below the line $\mathcal{L}$. So, because $h(z) - g(z)$ is monotone non-increasing due to Lemma 4, we should have $Z_l \leq z^{(1)} \leq z^{(2)} \leq Z_u$.

Using Lemma 2, the ADV's positive region is $(\mathcal{M}_0, \mathcal{M}_1, \mathcal{Q}_1, \mathcal{Q}_2, \mathcal{M}_2)$, where $\{\mathcal{M}_j\}_{j=1,2,3}$ and $\{\mathcal{Q}_j\}_{j=1,2}$ are as described in Lemma 2. The upper concave envelope conc-env$(\mathbf{r})$ upper-bounds the curve $\mathbf{r}$. Therefore, to show that curve $\mathbf{r}$ is entirely covered by the ADV's positive region, it is only enough to show its upper concave envelope conc-env$(\mathbf{r})$ is entirely covered (as can also be seen from Figure 2). Lets denote the line leaving $\mathcal{P}_j$ with slope one by $\mathcal{L}_j$ for $j = 1, 2$. The curve conc-env$(\mathbf{r})$ consists of three parts: the part above $\mathcal{L}_1$, the part below $\mathcal{L}_2$ and the part between $\mathcal{L}_1$ and $\mathcal{L}_2$ (the last part is indeed the line segment connecting $\mathcal{P}_1$ and $\mathcal{P}_2$). Interestingly, the line connecting $\mathcal{P}_1$ to $\mathcal{Q}_1$ and the line connecting $\mathcal{P}_2$ to $\mathcal{Q}_2$ both have slope 1. So, as it can be seen from Figure 2, if we show $\mathcal{Q}_1$ is above the line $h' = \beta$ and $\mathcal{Q}_2$ is to the right of the line $g' = \alpha$, then the conc-env$(\mathbf{r})$ will entirely be covered by the positive region and we are done. To see why this holds, first note that $\lambda$ has been picked so that $\mathcal{P} \triangleq (\mathcal{P}_g, \mathcal{P}_h) = \lambda \mathcal{P}_1 + (1 - \lambda)\mathcal{P}_2$. Due to Lemma 2,

$$\mathcal{Q}_1, h = \lambda(\frac{3}{2}h_1 + \frac{1}{2}g_1) + (1 - \lambda)(\frac{3}{2}h_2 + \frac{1}{2}g_2) = \frac{3}{2}\mathcal{P}_h + \frac{1}{2}\mathcal{P}_g$$

$$\mathcal{Q}_2, g = \lambda(\frac{3}{2}g_1 + \frac{1}{2}h_1) + (1 - \lambda)(\frac{3}{2}g_2 + \frac{1}{2}h_2) = \frac{3}{2}\mathcal{P}_g + \frac{1}{2}\mathcal{P}_h$$

Moreover, point $\mathcal{P} = (\mathcal{P}_g, \mathcal{P}_h)$ dominates the point $\mathcal{C} \triangleq (\frac{\alpha^2}{\alpha+\beta}, \frac{\beta^2}{\alpha+\beta})$ coordinate-wise. This dominance is simply true because points $\mathcal{C}$ and $\mathcal{P}$ are actually the intersections of the line $\mathcal{L} : h' - \beta = g' - \alpha$ (with slope one) with the line connecting $(0, \beta)$ to $(\alpha, 0)$ and with the curve `conc-env(r)` respectively. As `conc-env(r)` upper-bounds the line connecting $(0, \beta)$ to $(\alpha, 0)$, and because $\mathcal{L}$ has slope one, $\mathcal{P}_h \geq \mathcal{C}_h = \frac{\beta^2}{\alpha+\beta}$ and $\mathcal{P}_g \geq \mathcal{C}_g = \frac{\alpha^2}{\alpha+\beta}$. Putting all the pieces together,

$$\mathcal{Q}_{1,h} \geq \frac{3}{2}\frac{\beta^2}{\alpha+\beta} + \frac{1}{2}\frac{\alpha^2}{\alpha+\beta} = \frac{(\alpha^2 + \beta^2 - 2\alpha\beta) + 2\beta^2 + 2\alpha\beta}{2(\alpha+\beta)} = \beta + \frac{(\alpha-\beta)^2}{2(\alpha+\beta)} \geq \beta$$

$$\mathcal{Q}_{2,g} \geq \frac{3}{2}\frac{\alpha^2}{\alpha+\beta} + \frac{1}{2}\frac{\beta^2}{\alpha+\beta} = \frac{(\alpha^2 + \beta^2 - 2\alpha\beta) + 2\alpha^2 + 2\alpha\beta}{2(\alpha+\beta)} = \alpha + \frac{(\alpha-\beta)^2}{2(\alpha+\beta)} \geq \alpha$$

which implies $\mathcal{Q}_1$ is above the line $h' = \beta$ and $\mathcal{Q}_2$ is to the right of the line $g' = \alpha$, as desired. $\square$

## More details on polynomial implementation of Algorithm 1

---

**Algorithm 3:** Approximate One-Dimensional Optimization

---
**input**: function $f : [0,1] \to [0,1]$, additive error $\delta > 0$, Lipschitz Constant $C > 0$ ;
**output**: coordinate value $z \in [0,1]^n$ ;
Set $\epsilon \leftarrow \frac{\delta}{C}$ ;
Initialize $z^* \leftarrow 0$ ;
Initialize $z \leftarrow 0$ ;
**while** $z \leq 1$ **do**
    **if** $f(z) > f(z^*)$ **then**
        $z^* \leftarrow z$ ;
    $z \leftarrow z + \epsilon$ ;
**return** $z^*$

---

**Algorithm 4:** Approximate Annotated Upper-concave Envelope

---
**input**: function $f : [0,1] \to [0,1]$, function $g : [0,1] \to [0,1]$, additive error $\delta > 0$, Lipschitz Constant $C > 0$ ;
**output**: coordinate value $z \in [0,1]^n$ ;
Set $\epsilon \leftarrow \frac{\delta}{C}$ ;
Initialize stacks $s, t$ ;
Initialize $z \leftarrow 0$ ;
**while** $z \leq 1$ **do**
    **if** *s is empty or $f(z)$ is strictly larger than the first coordinate of the the top element of s* **then**
        **while** *s has at least two elements and the slope from (the second-to-top element of s) to (the top element of s) is less than the slope from (the top element of s) to $(f(z), g(z))$* **do**
            Pop the top element of $s$ ;
            Pop the top element of $t$ ;
        Push $(f(z), g(z))$ onto $s$ ;
        Push $z$ onto $t$ ;
    $z \leftarrow z + \epsilon$ ;
**return** $(s, t)$

---

## Analysis of the Binary-Search Bi-Greedy (proof of Theorem 2)

We start by the following technical lemma, which is used in various places of our analysis. The proof is immediate by `strong DR-SM` property (Definition 1).

**Lemma 5.** *For any $\mathbf{y}, \mathbf{z} \in [0,1]^n$ such that $\mathbf{y} \leq \mathbf{z}$, we have $\frac{\partial \mathcal{F}}{\partial x_i}(\mathbf{y}) - \frac{\partial \mathcal{F}}{\partial x_i}(\mathbf{z}) \geq 0, \forall i$.*

***Proof of Lemma 5.*** We rewrite this difference as a sum over integrals of the second derivatives:

$$\frac{\partial \mathcal{F}}{\partial x_i}(\mathbf{y}) - \frac{\partial \mathcal{F}}{\partial x_i}(\mathbf{z}) = \sum_{j=1}^{n} \left[ \begin{array}{l} \dfrac{\partial \mathcal{F}}{\partial x_i}(y_1, \ldots, y_{j-1}, y_j, z_{j+1}, \ldots, z_n) \\ -\dfrac{\partial \mathcal{F}}{\partial x_i}(y_1, \ldots, y_{j-1}, z_j, z_{j+1}, \ldots, z_n) \end{array} \right]$$

$$= \sum_{j=1}^{n} \int_{y_j}^{z_j} -\frac{\partial^2 \mathcal{F}}{\partial x_i \partial x_j}(y_1, \ldots, y_{j-1}, w, z_{j+1}, \ldots, z_n) dw \geq 0$$

To see why the last inequality holds, because of the `strong DR-SM` Proposition 3 implies that all of the second derivatives of $\mathcal{F}$ are always nonpositive. As $\forall i : z_i \geq y_i$, the RHS is nonnegative. □

***A modified zero-sum game.*** We follow the same approach and notations as in the proof of Theorem 1 (Section 2.1). Suppose $\mathbf{x}^*$ is the optimal solution. For each coordinate $i$ we again define a two-player zero-sum game between `ALG` and `ADV`, where the former plays $\hat{z}_i$ and the latter plays $x_i^*$. The payoff matrix for the `strong DR-SM` case, denoted by $\mathcal{V}_S^{(i)}(\hat{z}_i, x_i^*)$ is defined as before (Equation (1)); the only major difference is we redefine $h(.)$ and $g(.)$ to be the following functions,:

$$g(z) \triangleq \mathcal{F}(z, \mathbf{X}_{-i}^{(i-1)}) - \mathcal{F}(0, \mathbf{X}_{-i}^{(i-1)}) \ , \ h(z) \triangleq \mathcal{F}(z, \mathbf{Y}_{-i}^{(i-1)}) - \mathcal{F}(1, \mathbf{Y}_{-i}^{(i-1)}).$$

Now, similar to Lemma 1, we have a lemma that shows how to prove the desired approximation factor using the above zero-sum game. The proof is exactly as Lemma 1 and is omitted for brevity.

**Lemma 6.** *Suppose $\forall i \in [n] : \mathcal{V}_S^{(i)}(\hat{z}_i, x_i^*) \geq -\delta/n$ for constant $\delta > 0$. Then $2\mathcal{F}(\hat{\mathbf{z}}) \geq \mathcal{F}(\mathbf{x}^*) - \delta$.*

***Analyzing zero-sum games.*** We show that $\mathcal{V}_S^{(i)}(\hat{z}_i, x_i^*)$ is lower-bounded by a small constant, and then by using Lemma 6 we finish the proof. The formal proof, which appears in the supplementary materials, uses both ideas similar to those of Buchbinder et al. [2015], as well as new ideas on how to relate the algorithm's equilibrium condition to the value of the two-player zero-sum game.

**Proposition 4.** *if `ALG` plays the strategy $\hat{z}_i$ described in Algorithm 2, then $\mathcal{V}_S^{(i)}(\hat{z}_i, x_i^*) \geq -2C\epsilon/n$.*

***Proof of Proposition 4.*** Consider the easy case where $\frac{\partial \mathcal{F}}{\partial x_i}(0, \mathbf{X}_{-i}^{(i-1)}) \leq 0$ (and therefore we have $\frac{\partial \mathcal{F}}{\partial x_i}(0, \mathbf{Y}_{-i}^{(i-1)}) \leq 0$ due to `Strong DR-SM`). In this case, $\hat{z}_i = 0$ and hence $g(\hat{z}_i) = g(0) = 0$. Moreover, because of the `Strong DR-SM` property,

$$h(0) = \mathcal{F}(0, \mathbf{Y}_{-i}^{(i-1)}) - \mathcal{F}(1, \mathbf{Y}_{-i}^{(i-1)}) \geq -\frac{\partial \mathcal{F}}{\partial x_i}(0, \mathbf{Y}_{-i}^{(i-1)}) \geq 0,$$

$$h(x_i^*) - h(0) \leq g(x_i^*) - g(0) \leq x_i^* \cdot \frac{\partial \mathcal{F}}{\partial x_i}(0, \mathbf{X}_{-i}^{(i-1)}) \leq 0,$$

and therefore $\mathcal{V}_S^{(i)}(\hat{z}_i, x_i^*) = \frac{1}{2}g(0) + \frac{1}{2}h(0) - \max\left(g(x_i^*) - g(0), h(x_i^*) - h(0)\right) \geq 0$. The other easy case is when $\frac{\partial \mathcal{F}}{\partial x_i}(1, \mathbf{Y}_{-i}^{(i-1)}) \geq 0$ (and therefore $\frac{\partial \mathcal{F}}{\partial x_i}(1, \mathbf{X}_{-i}^{(i-1)}) \geq 0$, again because of `Strong DR-SM`). In this case $\hat{z}_i = 1$ and a similar proof shows $\mathcal{V}_S^{(i)}(1, x_i^*) \geq 0$.

The only remaining case (the not-so-easy one) is when $\frac{\partial \mathcal{F}}{\partial x_i}(0, \mathbf{X}_{-i}^{(i-1)}) > 0$ and $\frac{\partial \mathcal{F}}{\partial x_i}(1, \mathbf{Y}_{-i}^{(i-1)}) < 0$. In this case, Algorithm 2 runs the binary search and ends up at a point $\hat{z}_i$. Because of the monotonicity and continuity of the equilibrium condition of the binary search, there exists $\tilde{z}$ that is $(\epsilon/n)$-close to $\hat{z}_i$ and $\frac{\partial \mathcal{F}}{\partial x_i}(\tilde{z}, \mathbf{X}_{-i})(1 - \tilde{z}) + \frac{\partial \mathcal{F}}{\partial x_i}(\tilde{z}, \mathbf{Y}_{-i})\tilde{z} = 0$. By a straightforward calculation using the Lipschitz continuity of $\mathcal{F}$ with constant $C$ and knowing that $|\tilde{z} - \hat{z}_i| \leq \epsilon/n$, we have:

$$\mathcal{V}_S^{(i)}(\hat{z}_i, x_i^*) = \frac{1}{2}g(\hat{z}_i) + \frac{1}{2}h(\hat{z}_i) - \max\left(g(x_i^*) - g(\hat{z}_i), h(x_i^*) - h(\hat{z}_i)\right) \geq \mathcal{V}_S^{(i)}(\tilde{z}, x_i^*) - \frac{2C\epsilon}{n}$$

So, we only need to show $\mathcal{V}_S^{(i)}(\tilde{z}, x_i^*) \geq 0$. Let $\alpha \triangleq \frac{\partial \mathcal{F}}{\partial x_i}(\tilde{z}, \mathbf{X}_{-i}^{(i-1)})$ and $\beta \triangleq -\frac{\partial \mathcal{F}}{\partial x_i}(\tilde{z}, \mathbf{Y}_{-i}^{(i-1)})$. Because of Lemma 5, $\alpha + \beta \geq 0$. Moreover, $\alpha(1 - \tilde{z}) = \beta\tilde{z}$, and therefore we should have $\alpha \geq 0$ and $\beta \geq 0$. We now have two cases:

**Case 1 ($\tilde{z} \geq x_i^*$):**   $g(x_i^*) - g(\tilde{z}) \leq h(x_i^*) - h(\tilde{z})$ due to `strong DR-SM` and that $\tilde{z} \geq x_i^*$, so:

$$
\begin{aligned}
\mathcal{V}_S^{(i)}(\tilde{z}, x_i^*) &= \frac{1}{2}g(\tilde{z}) + \frac{1}{2}h(\tilde{z}) + (h(\tilde{z}) - h(x_i^*)) \\
&= \frac{1}{2}\int_0^{\tilde{z}} \frac{\partial \mathcal{F}}{\partial x_i}(x, \mathbf{X}_{-i}^{(i-1)})dx + \frac{1}{2}\int_{\tilde{z}}^1 -\frac{\partial \mathcal{F}}{\partial x_i}(x, \mathbf{Y}_{-i}^{(i-1)})dx + \int_{\tilde{z}}^{x_i^*} -\frac{\partial \mathcal{F}}{\partial x_i}(x, \mathbf{Y}_{-i}^{(i-1)}) \\
&\overset{(1)}{\geq} \frac{\tilde{z}}{2}\cdot\frac{\partial \mathcal{F}}{\partial x_i}(\tilde{z}, \mathbf{X}_{-i}^{(i-1)}) + \frac{(1-\tilde{z})}{2}\cdot\left(-\frac{\partial \mathcal{F}}{\partial x_i}(\tilde{z}, \mathbf{Y}_{-i}^{(i-1)})\right) + (x_i^* - \tilde{z})\left(-\frac{\partial \mathcal{F}}{\partial x_i}(\tilde{z}, \mathbf{Y}_{-i}^{(i-1)})\right) \\
&= \frac{\tilde{z}\alpha}{2} + \frac{(1-\tilde{z})\beta}{2} + (x_i^* - \tilde{z})\beta \\
&\overset{(2)}{\geq} \frac{\tilde{z}\alpha}{2} + \frac{(1-\tilde{z})\beta}{2} - \tilde{z}\beta \\
&\overset{(3)}{=} \frac{\alpha^2}{2(\alpha+\beta)} + \frac{\beta^2}{2(\alpha+\beta)} - \frac{\alpha\beta}{(\alpha+\beta)} = \frac{(\alpha-\beta)^2}{2(\alpha+\beta)} \geq 0,
\end{aligned}
$$

where inequality (1) holds due to the coordinate-wise concavity of $\mathcal{F}$, inequality (2) holds as $\beta \geq 0$ and $x_i^* \geq 0$, and equality (3) holds as $\beta\tilde{z} = \alpha(1-\tilde{z})$.

**Case 2 ($\tilde{z} < x_i^*$):**   This case is the reciprocal of Case 1, with a similar proof. Note that $g(x_i^*) - g(\tilde{z}) \geq h(x_i^*) - h(\tilde{z})$ due to `strong DR-SM` and the fact that $\tilde{z} < x_i^*$, so:

$$
\begin{aligned}
\mathcal{V}_S^{(i)}(\tilde{z}, x_i^*) &= \frac{1}{2}g(\tilde{z}) + \frac{1}{2}h(\tilde{z}) + (g(\tilde{z}) - g(x_i^*)) \\
&= \frac{1}{2}\int_0^{\tilde{z}} \frac{\partial \mathcal{F}}{\partial x_i}(x, \mathbf{X}_{-i}^{(i-1)})dx + \frac{1}{2}\int_{\tilde{z}}^1 -\frac{\partial \mathcal{F}}{\partial x_i}(x, \mathbf{Y}_{-i}^{(i-1)})dx + \int_{x_i^*}^{\tilde{z}} \frac{\partial \mathcal{F}}{\partial x_i}(x, \mathbf{X}_{-i}^{(i-1)}) \\
&\overset{(1)}{\geq} \frac{\tilde{z}}{2}\cdot\frac{\partial \mathcal{F}}{\partial x_i}(\tilde{z}, \mathbf{X}_{-i}^{(i-1)}) + \frac{(1-\tilde{z})}{2}\cdot\left(-\frac{\partial \mathcal{F}}{\partial x_i}(\tilde{z}, \mathbf{Y}_{-i}^{(i-1)})\right) + (\tilde{z} - x_i^*)\left(\frac{\partial \mathcal{F}}{\partial x_i}(\tilde{z}, \mathbf{X}_{-i}^{(i-1)})\right) \\
&= \frac{\tilde{z}\alpha}{2} + \frac{(1-\tilde{z})\beta}{2} + (\tilde{z} - x_i^*)\alpha \\
&\overset{(2)}{\geq} \frac{\tilde{z}\alpha}{2} + \frac{(1-\tilde{z})\beta}{2} + (\tilde{z} - 1)\alpha \\
&\overset{(3)}{=} \frac{\alpha^2}{2(\alpha+\beta)} + \frac{\beta^2}{2(\alpha+\beta)} - \frac{\alpha\beta}{(\alpha+\beta)} = \frac{(\alpha-\beta)^2}{2(\alpha+\beta)} \geq 0,
\end{aligned}
$$

where inequality (1) holds due to the coordinate-wise concavity of $\mathcal{F}$, inequality (2) holds as $\alpha \geq 0$ and $x_i^* \leq 1$, and equality (3) holds as $\beta\tilde{z} = \alpha(1-\tilde{z})$.    $\square$

Combining Proposition 4 and Lemma 6 for $\delta = 2C\epsilon$ finishes the analysis and the proof of Theorem 2.

## Detailed specifics of experiments in Section 4

### `Strong-DR` Non-concave Quadratic Programming (NQP)

We generated synthetic functions of the form $\mathcal{F}(\mathbf{x}) = \frac{1}{2}\mathbf{x}^T\mathbf{H}\mathbf{x} + \mathbf{h}^T\mathbf{x} + c$. We generated $\mathbf{H} \in \mathbb{R}^{n\times n}$ as a matrix with every entry *uniformly distributed* in $[-1, 0]$, and then symmetrized $\mathbf{H}$. We then generated $\mathbf{h} \in \mathbb{R}^n$ as a vector with every entry uniformly distributed in $[0, +1]$. Finally, we solved for the value of $c$ to make $\mathcal{F}(\vec{0}) + \mathcal{F}(\vec{1}) = 0$.

### `Weak-DR` Non-concave Quadratic Programming (NQP)

This experiment is the same as in the previous subsection, except that the diagonal entries of $\mathbf{H}$ are uniformly distributed in $[0, +1]$ instead, making the resulting function $\mathcal{F}(\mathbf{x})$ only `weak DR-SM` instead.

**Softmax extension of Determinantal Point Processes (DPP)**

We generated synthetic functions of the form $\mathcal{F}(\mathbf{x}) = \log \det(\text{diag}(\mathbf{x})(\mathbf{L} - \mathbf{I}) + \mathbf{I})$, where $\mathbf{L}$ needs to be positive semidefinite. We generated $\mathbf{L}$ in the following way. First, we generate each of the $n$ eigenvalues by drawing a uniformly random number in $[-0.5, 1.0]$ and using $e$ raised to that number. This yields a diagonal matrix $\mathbf{D}$. We then generate a random unitary matrix $\mathbf{V}$ and then set $\mathbf{L} = \mathbf{V}\mathbf{D}\mathbf{V}^T$. By construction, $\mathbf{L}$ is positive semidefinite and has the specified eigenvalues.

## Geometric proof via picture- larger versions of Figure 2 and Figure 1.

Here are the larger versions of Figure 2 and Figure 1.

## More application domain details

Here is a list containing further details about applications in machine learning, electrical engineering and other application domains.

**Special Class of Non-Concave Quadratic Programming (NQP).**

- The objective is to maximize $\mathcal{F}(\mathbf{x}) = \frac{1}{2}\mathbf{x}^T\mathbf{H}\mathbf{x} + \mathbf{h}^T\mathbf{x} + c$, where off-diagonal entries of $\mathbf{H}$ are non-positive (and hence these functions are `Weak DR-SM`).

- Minimization of this function (or equivalently maximization of this function when off-diagonal entries of $\mathbf{H}$ are non-negative) have been studied in Kim and Kojima [2003] and Luo et al. [2010], and has applications in communication systems and detection in MIMO channels [Luo et al., 2010].

- Another application of quadratic submodular optimization is large-scale price optimization on the basis of demand forecasting models, which has been studied in Ito and Fujimaki [2016]. They show the price optimization problem is indeed an instance of weak-DR minimization.

**Revenue Maximization over Social Networks.**

- The model was proposed in Bian et al. [2017b] and is a generalization of the revenue maximization problem addressed in Hartline et al. [2008].

- A seller wishes to sell a product to a social network of buyers. We consider restricted seller strategies which freely give (possibly fractional) trial products to buyers: this fractional assignment is our input $\mathbf{x}$ of interest.

- The objective takes two effects into account: (i) the revenue gain from buyers who didn't receive free product, where the revenue function for each such buyer is a nonnegative nondecreasing `Weak DR-SM` function and (ii) the revenue loss from those who received free product, where the revenue function for each such buyer is a nonpositive nonincreasing `Weak DR-SM` function. The combination for all buyers is a nonmonotone `Weak DR-SM` function and additionally is nonnegative at $\vec{0}$ and $\vec{1}$.

**Map Inference for Determinantal Point Processes (DPP) & Its Softmax-Extension.**

- DPP are probabilistic models that arise in statistical physics and random matrix theory, and their applications in machine learning have been recently explored, e.g. [Kulesza et al., 2012].

- DPPs can be used as generative models in applications such as text summarization, human pose estimation, or news threading tasks [Kulesza et al., 2012].

- A discrete DPP is a distribution over sets, where $p(S) \sim \det(A_S)$ for a given PSD matrix $A$. The log-likelihood estimation task corresponds to picking a set $\hat{S} \in \mathcal{P}$(feasible set, e.g. a matching) that maximizes $f(S) = \log(\det(A_S))$. This function is non-monotone and submodular. Note that as a technical condition to apply bi-greedy algorithms, we require that $\det(A) \geq 1$ (implying $f(\vec{1}) \geq 0$).

- The approximation question was studied in [Gillenwater et al., 2012]. Their idea is to first find a fractional solution for a continuous extension (hence a a continuous submodular maximization step is required) and then rounding the solution. However, they sometimes need a fractional solution in $\text{conv}(\mathcal{P})$ (so, the optimization task sometimes fall out of the hypercube, making rounding more complicated).

- Beyond multilinear extension, the other continuous extension that has been used in this literature is called the *softmax extension* [Gillenwater et al., 2012, Bian et al., 2017a]:

$$\mathcal{F}(\mathbf{x}) = \log \mathbf{E}_{S \sim \mathcal{I}_\mathbf{x}}[\exp(f(S))] = \log \det\left(\text{diag}(\mathbf{x})(A - I) + I\right)$$

where $\mathcal{I}_\mathbf{x}$ is the independent distribution with marginals $\mathbf{x}$ (i.e. each item $i$ is independently in the set w.p. $x_i$).

- $\mathcal{F}(\mathbf{x})$ is `Strong DR-SM` and non-monotone [Bian et al., 2017a]. In almost all machine learning applications, the rounding works on an unrestricted problem. Hence the optimization that needs to be done is `Strong DR-SM` optimization over unit hypercube.
- One can think of adding a regularizer term $\lambda \|\mathbf{x}\|^2$ to the log-likelihood objective function to avoid overfitting. In that case, the underlying fractional problem becomes a `Weak DR-SM` optimization over the unit hypercube when $\lambda$ is large enough.

**Log-Submodularity and Mean-Field Inference.**

- Another probabilistic model that generalizes DPP and all other strong Rayleigh measures [Li et al., 2016, Zhang et al., 2015] is the class of *log-submodular* distributions over sets, i.e. $p(S) \sim \exp(f(S))$ where $f(\cdot)$ is a discrete submodular functions. MAP inference over this distribution has applications in machine learning and beyond [Djolonga and Krause, 2014].
- One variational approach towards this MAP inference task is to do *mean-field inference* to approximate the distribution $p$ with a product distribution $\mathbf{x} \in [0,1]^n$, i.e. finding $\mathbf{x}^*$ that:

$$\mathbf{x}^* \in \underset{\mathbf{x} \in [0,1]^n}{\operatorname{argmax}} \ \mathbb{H}(\mathbf{x}) - \mathbf{E}_{S \sim \mathcal{I}_x}[\log p(S)] = \underset{\mathbf{x} \in [0,1]^n}{\operatorname{argmin}} \ \mathrm{KL}(\mathbf{x} \| p)$$

where $\mathrm{KL}(\mathbf{x} \| p) = \mathbf{E}_{S \sim \mathcal{I}}[\frac{\log \mathcal{I}_x(S)}{\log p(S)}]$.

- The function $\mathcal{F}(\mathbf{x}) = \mathbb{H}(\mathbf{x}) - \mathbf{E}_{S \sim \mathcal{I}_x}[\log p(S)]$ is `Strong DR-SM` [Bian et al., 2017a].

**Cone Extension of Continuous Submodular Maximization.**

- Suppose $\mathcal{K}$ is a proper cone. By considering the lattice corresponding to this cone one can generalize DR submodularity to $\mathcal{K}$-DR submodularity [Bian et al., 2017a].
- An interesting application of this cone generalization is minimizing the loss in the logistic regression model with a particular non-separable and non-convex regularizer, as described in [Antoniadis et al., 2011, Bian et al., 2017a]. Bian et al. [2017a] show the vanilla version is a $\mathcal{K}$-`Strong DR-SM` function maximization for some particular cone.
- Note that by adding a $\mathcal{K}$-$\ell_2$-norm regularizer $\lambda \|\mathbf{A}\mathbf{x}\|^2$, the function will become `Weak DR-SM`, where $\mathbf{A}$ is a matrix with generators of $\mathcal{K}$ as its column. Here is the logistic loss:

$$l(\mathbf{x}, \{y_t\}) = \frac{1}{T} \sum_{t=1}^{T} f_t(\mathbf{x}, y_t) = \frac{1}{T} \sum_{t=1}^{T} \log\left(1 + \exp\left(-y_t \mathbf{x}^T \mathbf{z}^t\right)\right)$$

where $y_t$ is the label of the $t^{\text{th}}$ data-point, $\mathbf{x}$ are the model parameters, and $\{\mathbf{z}^t\}$ are feature vectors of the data-points.

**Remark 1.** In many machine learning applications, and in particular MAP inference of DPPs and log-submodular distributions, unless we impose some technical assumptions, the underlying `Strong DR-SM` (or `Weak DR-SM`) function is *not* necessarily positive (or may not even satisfy the weaker yet sufficient condition $\mathcal{F}(\vec{0}) + \mathcal{F}(\vec{1}) \geq 0$). In those cases, adding a positive constant to the function can fix the issue, but the multiplicative approximation guarantee becomes weaker. However, this trick tends to work in practice since these algorithms tend to be near optimal.