[Reviews · NeurIPS 2018]

Reviewer 1



The paper is about maximizing non-monotone continuous submodular functions on the hypercube [0,1]^n. The main result of the paper is an algorithm that provides a tight (1/2-eps)-optimal solution using O(n^2/eps) complexity. The algorithm is built on the double-greedy algorithm proposed by Buchbinder et al for maximizing non-monotone submodular set functions, however, it turns out that a number of novel ideas have to be added to extend the algorithm to continuous submodular functions (e.g. the coordinate-wise zero-sum game strategy to decide the value of the final solution at each coordinate). A 1/2-optimal algorithm was already known for DR-submodular functions and a 1/3-opt efficient algorithm was known for continuous submodular functions, both based on the double-greedy method, however, I find the case of general continuous submodular functions much more difficult than maximizing non-monotone DR-submodular functions. All in all, after a careful read, I find the results of the paper quite novel and I recommend acceptance. I don't have any major comments and only have a few suggestions. First of all, the dependency to the dimension n, i.e. n^2, can perhaps be improved by a more intelligent (and adaptive over rounds using submodularity and lazy updates) discretization method. Second, it's worth to mention how many function computations your algorithm needs (i.e. is it O(n/eps) or O(n^2/eps)--I suspect that the latter is the case ).

Reviewer 2



The paper gives a tight 1/2 approximation for maximizing a continuous submodular function subject to a box constraint i.e. max f(x) subject to a_i <= x_i <= b_i where f is a continuous submodular function. The algorithm and proof are inspired by the result of Buchbinder et al. for discrete submodular functions. The overall structure of the algorithm and the proof are similar to the previous work (the algorithm works coordinate by coordinate and the proof tracks similar quantities). The main new difficulty here is in dealing with the 1 dimensional problem and showing that no matter what the optimal value is, the gain of the algorithm is always at least 1/2 the drop in the optimal value after fixing a coordinate. The proof is quite technical and I have not checked it carefully. A comment on the illustration: by lemma 4 in the supplement, h(z)-g(z) is monotone so the curve cannot really behave wildly like the figure shows. Maybe including that statement and correct the drawing in the main body would help the reader's intuition.

Reviewer 3



The paper proposes an 1/2-approximation algorithm for non-monotone continuous submodular maximization problem. This is the optimal approximation ratio. I checked the proof outline in the paper, and skimmed them in supplementary material. I did not find any serious flaw. I believe this paper should be accepted. The obtained result is very strong, and the proof technique (reduction to a zero-sum game) is quite novel. Even the practical performance, shown in Table 1, is not so different with the existing methods, it has a solid theoretical strength. [minor] - Are the differences in Table 1 significant? (i.e., could you show the variance?)